# Controlling Trophoblast Cell Fusion in the Human Placenta—Transcriptional Regulation of Suppressyn, an Endogenous Inhibitor of Syncytin-1

**DOI:** 10.3390/biom13111627

**Published:** 2023-11-07

**Authors:** Jun Sugimoto, Danny J. Schust, Makiko Sugimoto, Yoshihiro Jinno, Yoshiki Kudo

**Affiliations:** 1Department of Obstetrics and Gynecology, Graduate School of Medical Sciences, Hiroshima University, Hiroshima 734-8551, Japanyoshkudo@hiroshima-u.ac.jp (Y.K.); 2Department of Obstetrics and Gynecology, Duke University, Durham, NC 27710, USA; 3Department of Molecular Biology, University of the Ryukyus, Okinawa 903-0215, Japan

**Keywords:** suppressyn, syncytin, HERV, placenta, cell fusion, promoter, DNA methylation, hypoxia inducible factor (HIF)

## Abstract

Cell fusion in the placenta is tightly regulated. Suppressyn is a human placental endogenous retroviral protein that inhibits the profusogenic activities of another well-described endogenous retroviral protein, syncytin-1. In this study, we aimed to elucidate the mechanisms underlying suppressyn’s placenta-specific expression. We identified the promoter region and a novel enhancer region for the gene encoding suppressyn, *ERVH48-1*, and examined their regulation via DNA methylation and their responses to changes in the oxygen concentration. Like other endogenous retroviral genes, the *ERVH48-1* promoter sequence is found within a characteristic retroviral 5′ LTR sequence. The novel enhancer sequence we describe here is downstream of this LTR sequence (designated EIEs: ERV internal enhancer sequence) and governs placental expression. The placenta-specific expression of *ERVH48-1* is tightly controlled by DNA methylation and further regulated by oxygen concentration-dependent, hypoxia-induced transcription factors (HIF1α and HIF2α). Our findings highlight the involvement of (1) tissue specificity through DNA methylation, (2) expression specificity through placenta-specific enhancer regions, and (3) the regulation of suppressyn expression in differing oxygen conditions by HIF1α and HIF2α. We suggest that these regulatory mechanisms are central to normal and abnormal placental development, including the development of disorders of pregnancy involving altered oxygenation, such as preeclampsia, pregnancy-induced hypertension, and fetal growth restriction.

## 1. Introduction

Human endogenous retroviruses (HERVs) are remnants of ancient retroviral infections that have become integrated into the human genome. HERVs are believed to have originated from exogenous retroviral infections that occurred millions of years ago [1,2,3,4]. Through a process called retrotransposition, these viruses integrated their genetic materials into the germ cells of our ancestors, allowing for their sequences to be inherited across generations. While the majority of HERVs have accumulated mutations over time that render them incapable of producing infectious particles, some have retained the potential for the transcription and translation of segments of retroviral proteins (ex: the env region of suppressyn [5]) that are regulated by traditional 5′ and 3′ LTR (long terminal repeat) sequences. The 5′ and 3′ LTRs consist of unique U3 and U5 regions and regulatory regions (R), including transcriptional start sites (TSSs). In addition to the core polymerase II promoter elements (e.g., TATA box), LTRs often harbor TFBSs (transcription factor binding sites) and may contain splice donor (SD) sites within the U5 region. These elements can be induced under various physiological or pathological conditions as promoter and/or enhancer sequences [1,6]. Epigenetic modifications of these LTRs and neighboring sequences have been reported [7,8]. In the human placenta, previous studies have shown that the tissue-specific expression of an important profusogenic HERV protein, syncytin-1 [9,10,11], is regulated via DNA methylation within the gene promoter region [7,8,12,13]. Our general understanding of endogenous retrovirus gene expression remains limited and may include mechanisms that are more complex than simple LTR regulation or DNA methylation. As normal and abnormal HERV gene expression and HERV protein production are linked to both beneficial and detrimental effects, elucidating the regulation of HERV expression may be particularly important for our understanding of tissue development and dysfunction, specifically in the placenta.

The hemochorial villous human placenta is comprised of two primary trophoblast subtypes, namely cytotrophoblast cells (CTB) and the multinucleated syncytiotrophoblast (STB) [14]. Proliferative progenitor cytotrophoblast cells follow two distinct pathways of differentiation. Some acquire an invasive phenotype and exit the villous placenta as extravillous cytotrophoblast cells (EVT) [15]. These EVT cells traverse the maternal decidua, where they encounter a unique population of maternal immune cells [16]. An important subpopulation of EVT will invade and remodel the maternal spiral arteries, where they are called endovascular trophoblast cells, and some may ultimately differentiate into the trophoblast giant cells observed at the interface between the endometrium and myometrium and within the inner third of the myometrium [15,17,18]. In contrast, other proliferative villous cytotrophoblast cells cease cell division and merge to generate the villous syncytiotrophoblast (STB), a single multinucleated cell that covers the entire surface of the villous human placenta [19].

Oxygen levels in the human placenta vary quite significantly across gestation during normal development [20]. Remarkably, the physiologic oxygen concentrations during much of the first trimester of gestation are very low (approximately 1–3% O_2_), largely because the endovascular EVTs plug the distal ends of the maternal uterine spiral arteries [21,22]. The resulting low oxygen environment supports the proliferation of CTBs and their differentiation to immature EVTs, a process that is most robust during the early stages of placental development. The EVT plugs within the spiral arteries break down near the end of the first trimester, allowing for the free flow of maternal blood into the intervillous space. This process increases the physiologic oxygen levels in the placenta, but only to about 8% O_2,_ and allows for the full differentiation of EVTs [21,22,23]. Most, if not all, mammalian cells utilize a transcription factor complex named hypoxia inducible factor (HIF) to mediate responses to low oxygen levels. This complex includes two major products: the HIF alpha (α) subunit and the HIF beta (β) subunit. The latter subunit is also called the aryl hydrocarbon receptor nuclear transfer factor (ARNT), and the former consists of three isoforms (HIF1α, HIF2α, and HIF3α). The human placenta is known to endogenously express HIF1α and HIF2α at high levels [24,25].

We identified and reported the presence of an endogenous inhibitor of cytotrophoblast cell fusion that is also derived from the env of an HERV [5]. Based on its mechanism of action in the placenta, we named this cell fusion inhibitor suppressyn; it is derived from an endogenous retrovirus in the H family and its gene symbol is *ERVH48-1* (HGNC:17216). We demonstrated that suppressyn binds to the receptor for *ERVW-1* (syncytin-1), which is the widely distributed neutral amino acid transporter known as alanine serine cysteine transporter 2 (ASCT2), also referred to as *SLC1A5* [10]. This binding event leads to an inhibition of the normal fusogenic effects mediated by the binding of syncytin-1 and ASCT2 [5,26,27]. *ERVH48-1* expression is largely limited to the human placenta; suppressyn translation products have been detected specifically in the placenta, where they can be localized in CTB, in cytotrophoblast cells within cell columns, and in extravillous cytotrophoblasts (EVT) [4,5,11,26,27,28]. Experimental systems using primary cultured trophoblast cells have confirmed that suppressyn levels are oxygen-sensitive, with increases in both *ERVH48-1* transcription and suppressyn translation in response to low oxygen conditions [26].

Syncytialization in the human placenta is essential to many of its central functions, including intercommunication between the mother and fetus through gas exchange, nutrient transport and hormone production, and protection from infection. The process of cell fusion is therefore tightly regulated, and dysregulation has been associated with disorders such as preeclampsia [13,29,30]. Since suppressyn directly inhibits the profusogenic effects of syncytin-1, we were interested in the mechanisms that regulate suppressyn levels in the placenta, as a disruption of its expression could directly impact placental formation and potentially lead to the development of various perinatal disorders related to abnormal placentation. Of particular interest to us were those regulatory mechanisms involved in suppressyn’s tissue specificity and in gene and protein expression changes in response to local oxygen conditions. 

## 2. Methods

### 2.1. Cell Cultures

BeWo, JEG3, and HeLa cells were purchased from ATCC (Manassas, VA, USA). BeWo and JEG3 cells were cultured in Ham’s F12 (087-08335: Fuji Film, Tokyo, Japan) supplemented with 10% FBS (Fetal Bovine Serum; Gibco-Thermo Fisher Scientific, Waltham, MA, USA) at 37 °C in humidified 2% CO_2_ and 20% O_2_. HTR-8/SVneo cells (referred to as HTR8 cells throughout for simplicity) [31] were a kind gift from Professor Charles Graham of the Department of Anatomy and Cell Biology at Queen’s University, Kingston, ON, Canada. The endometrial Ishikawa cell line was generously supplied by Dr. Susan Nagel at the University of Missouri. HTR8, Ishikawa, and Hela cells were cultured in DMEM (041-29775; Fuji Film, Tokyo, Japan) supplemented with 10% FBS at 37 °C in humidified 2% CO_2_ and 20% O_2_. 

### 2.2. Vectors and Constructs

Promoter and enhancer sequences were PCR-amplified using genomic DNA from term placenta. High-fidelity PCR reagents (KOD One PCR master mix; KMM-101 Toyobo, Osaka, Japan) were used for 35 cycles of first-round PCR. A second round of nested PCR was then performed using restriction enzyme site-linked primers (Appendix A). After purification using the FastGene gel/PCR extraction kit (FG-91202; NIPPON Genetics, Tokyo, Japan), PCR products were digested with *Kpn*I and *Xho*I restriction enzymes (310-00212, 312-00392; NIPPON GENE, Toyama, Japan) and cloned into the pGL3-basic (*Kpn*I/*Xh*oI) vector (E175A; Promega, Madison, WI, USA). Clones were sequenced with BigDye Terminator sequencing, and those with 100% sequence identity to the reference sequence were used for transfection.

HIF1α, HIF2α, and ARNT were PCR-amplified using a reverse-transcribed RNA product from BeWo cells. Following 35 cycles of first-round amplification, *Eco*RV and *Bam*HI site-linked primers (Appendix A) were used for nested PCR. PCR products were digested with *Eco*RV/*Bam*HI (317-00462, 315-00061; NIPPON GENE, Toyama, Japan) and cloned into the pFlag-EF1 vector [5] that produced a flag-tag-fused protein. 

### 2.3. Mutagenesis

Primers for deletional mutagenesis are listed in Appendix A. Twenty nanograms of original plasmid was used for amplification with the high-fidelity KOD One polymerase (KMM-101; Toyobo, Osaka, Japan) as per the manufacturer’s instructions. PCR products were used to directly transform DH5α competent cells, and plasmid DNA was extracted from several clones for sequencing. Plasmids with a 100% sequence match were identified and used for transfection.

### 2.4. Transient Transfections and Luciferase Assays

A total of 2 × 10^4^ cells from individual trophoblast, endometrial and cervical cell lines were used to seed 48-well plates and incubated for 24 h prior to plasmid transfection. Cells were co-transfected with 200 ng of pGL3 firefly luciferase plasmid and 10 ng of pRL-TK renilla luciferase plasmid (E2241; Promega, Madison, WI, USA) using Lipofectamine 2000 (11668027; Life Technologies, Carlsbad, CA, USA) for internal standardization. All transfection experiments were performed in duplicate. Cells were lysed 24 h after transfection with passive lysis buffer (E1910; Promega, Madison, WI, USA). Firefly and renilla luciferase activities were measured using a GloMax Discover Microplate Reader (GM3000; Promega, Madison, WI, USA). Promoter activities were normalized to renila luciferase activities. Co-transfection of hypoxia-related genes were conducted as follows: 2 × 10^4^ cells for each individual cell line were incubated in appropriate media in 48-well plates for 24 h, and then 200 ng of pGL3 plasmid, 250 ng of pFlag plasmid, and 10 ng of pRL-TK were transfected using 0.5 μL of Lipofectamine 2000 according to the manufacturer’s instructions. Detection of luciferase activities was analyzed as above. 

### 2.5. Bisulfite Methylation Analysis

Sodium bisulfite treatment was performed using an EZ DNA methylation-lightning kit (D5030T; Zymo Research, Irvine, CA, USA) as per the manufacturer’s instructions. Two micrograms of genomic DNA was used in each experiment, and the bisulfite-treated DNA was amplified via KAPATaq (KK1024; Kapa biosystems, Wilmington, MA, USA) using the following conditions: denaturation at 95 °C for 3 min, followed by 40 cycles of 95 °C for 30 s, annealing at 55 °C for 30 s and 72 °C for 1 min, and a final elongation phase at 72 °C for 3 min. All primer sequences are shown in Appendix A. These PCR products were cloned into the pGEM-T easy vector (A137A; Promega, Madison, WI, USA), and colony-specific PCR was performed for >10 clones each using vector primers (M13 forward and M13 reverse). After conventional PCR product purifications using the FastGene gel/PCR extraction kit (FG-91202; Nippon Genetics, Tokyo, Japan), PCR products were directly sequenced with the M13 reverse primer using a BigDye Terminator v1.1cycle sequencing kit (4337452; Applied Biosystems, Waltham, MA, USA). 

### 2.6. Statistical Analyses

Data are expressed and plotted as means ± standard deviations. Means were compared using Mann–Whitney U tests. Statistical significance was defined as * *p* < 0.05 or ** *p* < 0.01, as indicated. 

## 3. Results

### 3.1. Promoter Activity of the ERVH48-1 5′ LTR Genome Sequence

To better define the promoter sequence of suppressyn, we focused on the 5′ LTR of *ERVH48-1*, a transcriptional regulatory sequence that is characteristic of endogenous retroviruses, and verified its activity as a promoter. The total length of the 5′ LTR of suppressyn is 360 bp, which comprises a U3 segment of 246 bp, an R segment of 65 bp, and a U5 segment of 59 bp (Figure 1); the transcript (Transcript (+1) of the NM_001308491.2) begins in the R region downstream from a TATA box located at the U3 region (+30 bp). The 3′ flanking sequence is tandemly flanked by upstream MERA41 and AluY sequences. Twenty-eight CpG sites are present in this region and are expected to be targets for DNA methylation (Figure 1).

We amplified the 5′ LTR region using primers with restriction enzyme (*Kpn*I/*Xho*I) linkers (Appendix A) and cloned this product into the *Kpn*I/*Xho*I site of the pGL3-basic vector. Promega’s dual-luciferase assay was performed using these constructs and confirmed minimal promoter activity in all cell lines (BeWo, JEG3, HTR8, Ishikawa, and HeLa) transfected with this modified vector. Over 20-fold lower activity was noted when compared to similar experiments in the same cells using the promoter sequence of endogenous retrovirus-derived syncytin-1 (Appendix A). This suggested the presence of an enhancer sequence in the vicinity of the MER-A41, AluY sequences, and/or the 5′ flanking sequence (EH38E2145129), which was further predicted by the ENCODE registry of candidate cis-regulatory elements (cCREs) analysis of the sequence upstream of the 5′ LTR (Figure 2). Each of the modified sequences was cloned via PCR amplification (Appendix A) and introduced into BeWo, JEG3, HTR8, Ishikawa, and HeLa cells to allow for the assessment of the enhancer effects of each of these sequences. The 5′ flanking sequence (pGL3-5Enh) alone did not display promoter activity, indicating that the remainder of our sequences were expected to function as enhancers. A weak enhancer effect (*p* < 0.05) was observed for the sequence containing the AluY sequence (pGL3-Alu-LTR), but no statistically significant enhancement of expression was observed for the candidate enhancer sequence (pGL3-5Enh-LTR) or MERA41 (pGL3-MER-LTR) (Figure 2). Further, the GCM1 binding sequence that functions as a major site for tissue-specific transcriptional control for *ERVW-1* is not found in the *ERVH48-1* LTR and 5′ flanking sequences. Thus, unlike syncytin-1, the placental expression of suppressyn is not dependent on GCM1, which is only expressed in trophoblast cells [32,33,34] (Appendix A).

### 3.2. DNA Methylation Analysis of the 5′ LTR and Upstream Enhancer Sequences

Next, a DNA methylation analysis of the CpG dinucleotides of these promoter and enhancer sequences was carried out with the aim of determining an epigenetic mechanism that regulates the tissue-specific expression of suppressyn [5,11,26,27,28]. The genomic DNA from all cell lines used for the promoter analysis (the choriocarcinoma cell lines, BeWo and JEG3; the primary culture-derived trophoblast cell line, HTR8; the endometrial adenocarcinoma-derived cell line, Ishikawa; and the cervical cancer-derived cell line, HeLa) was subjected to bisulfite sequencing. Of the 28 total CpG sites within the sequence spanning the 5′ flanking region to the 5′ LTR, only 10 are located in the 5′ LTR region (Figure 1). The bisulfite sequencing revealed relatively low to absent CpG methylation levels within the 5′ LTR and 5′ flanking sequences for both of the trophoblast cell lines that are known to express suppressyn (BeWo and JEG3), but revealed almost universal DNA methylation in these regions for the cell lines that have absent suppressyn expression (Ishikawa, HeLa, and the trophoblast cell line, HTR8). More specifically, the 10 CpG sites within the 5′ LTR were methylated at the following levels: BeWo (0%), JEG3 (5%), Ishikawa (91%), HeLa (96%), and HTR8 (97%) (Figure 3). Similarly, the DNA methylation levels in the 5′ LTR of *ERVH48-1* in the genomic DNA from four human term placentas were very low, while these CpG sites were heavily methylated in the pancreas and thyroid tissues, neither of which express suppressyn (Appendix A). In summary, DNA methylation within the 5′ LTR promoter region of *ERVH48-1* appears to be at least one of several regulatory mechanisms underlying the placenta-specific expression of suppressyn that we previously demonstrated at the transcriptional level using Northern blots [28] and RT-PCR [5,11] and at the protein level using Western blots [5,27] and immunohistochemistry [5,26,27]. These results were confirmed at the level of histone modification using an in silico analysis. We assessed the sequence in and around the *ERVH48-1* coding region on chr 21 for DNA methylation and assessed the histone modification status using the CHIP-Atlas web program (https://chip-atlas.org (accessed on 1 April 2022)) [35]. The in silico data showed that many tissues (e.g., liver, lung, kidney, breast, uterus, and pancreas) were highly methylated throughout the *ERVH48-1* gene sequence; the 5′ LTR and EIE loci associated with ERVH48-1 were notable exceptions (Appendix A). The histone CHIP analyses available from this same source showed that H3K4me3 and H3K27ac binding sites were also predicted in the *ERVH48-1* 5′ LTR and EIE loci (Appendix A) [4,35].

### 3.3. Placenta-Specific Enhancer Sequences: EIEs

We also searched for novel sequences that control the *ERVH48-1* promoter and focused on candidate enhancer sequences predicted by the ENCODE depository: cCREs (EH38E2145125, EH38E2145126, and EH38E2145127). A region 2072 bp downstream from the transcription start point (+1) was cloned using PCR-based amplification (Appendix A), and its regulatory activity was analyzed via dual-luciferase assay. Putative enhancer sequences were designated as ERV Internal Enhancer sequences (EIEs). Our demonstration of absent promoter activity for the EIE sequence alone (pGL3-EIE) strongly suggested that this sequence exerted enhancer activity. Interestingly, only the suppressyn-expressing BeWo and JEG3 cells showed significant enhancements of their luciferase activities when this sequence was combined with the 5′ LTR (pGL3-LTR-EIE) (Figure 4a). In comparison, the enhancer sequence derived from the 5′ flanking sequence (pGL3-5Enh-LTR-EIE) in the BeWo and JEG3 cells had no effect (no significant differences were identified compared with pGL3-LTR-EIE) on the promoter activity, including the EIEs. We next sought to identify the sequence responsible for these enhancer effects by creating constructs that deleted this EIE sequence using restriction enzyme sites (*Sma*I at 838 bp, *Eco*RV at 1188 bp, and *Bgl*II at 1895 bp) surrounding the enhancer (sequence predicted in the cCREs was deleted, as shown in Figure 4b). We found that the introduction of the deletion by *Bgl*II (pGL3-LTR-EIE Del3) significantly reduced the luciferase activity to about half of that of the EIEs (pGL3-LTR-EIE) alone. Luciferase activity was completely lost for the deletion constructs created by *Sma*I (pGL3-LTR-EIE Del1) and *Eco*RV (pGL3-LTR-EIE Del2). Since the activities were less than that of the 5′ LTR-only sequence, we surmised that the remaining sequences deleted in these constructs exert transcriptional suppressor activities. While *ERVH48-1* did not have the characteristic binding site for GCM1 that was seen for *ERVW-1,* the transcription binding site prediction program, TFBIND (software for searching transcription factor binding sites, https://tfbind.hgc.jp, 1 April 2022 [36]), showed that GATA transcription factors are predicted to bind to two *ERVH48-1* EIEs sites, as well as a site within the *ERVH48-1* 5′ LTR (Figure 4b). Members of this group of transcription factors are only found to be expressed in the trophoblast cell lines (BeWo and JEG3) [37,38,39] that express suppressyn, but not the lines, including one trophoblast cell line, that do not express this anti-fusogenic protein (HTR8, Ishikawa, and HeLa) (Appendix A). We again used in silico analyses (CHIP-Atlas programs (https://chip-atlas.org (accessed on 1 April 2022))) [35] to identify transcription factor binding potential. Such analyses predicted GATA2 binding in the *ERVH48-1* 5′ LTR only in placental samples and predicted GATA binding in the EIE regions in other tissue samples (Appendix A) [4,35]. Since the deletion of GATA binding sites within those EIEs outside of the 5′ LTR appears to negatively regulate the 5′ LTR, the 1188 bp to 2318 bp EIE sequences may act as core enhancer sites at which GATA may exert an influence. This is consistent with the hypothesis that placenta-specific enhancer effects may be related to the presence or absence of GATA transcription factors, and furthermore, that the loss of enhancement with these mutant constructs may result directly from a deficiency in GATA transcription factor binding sites.

### 3.4. Oxygen-Related Changes in ERVH48-1 Promoter Activity

The effects of the surrounding oxygen conditions on *ERVH48-1* regulatory 5′ LTR and LTR-EIE activities and any mechanisms underlying these effects remain incompletely studied. Here, luciferase reporter constructs containing the *ERVH48-1* promoter and enhancer sequences were cultured under 2% and 20% oxygen concentrations to mimic low to physiologic (depending on trimester) and supraphysiologic oxygen concentrations, respectively. This particular high oxygen condition was chosen to allow for a comparison to the levels that are used frequently in the extant literature as supraphysiologic levels. For the cell lines that lack suppressyn expression (HTR8, Ishikawa, and HeLa cells), the luciferase activities were low and responded little to the changes in oxygen (Figure 5a). In contrast, enhanced luciferase activities were seen in the BeWo and JEG3 cells in response to low (2%) oxygen. While the BeWo cells were limited to a significant increase in the luciferase activity only for the LTR construct (pGL3-LTR), there was a more than two-fold increase in the luciferase activity in the presence of 2% oxygen for all constructs that were introduced into the JEG3 cells. 

An HIF-binding sequence (-cacgt-; Figure 1) can be identified in the 5′ LTR sequence of suppressyn in all of the cell lines used in this study, and with the notable exceptions of HIF2α in the Ishikawa cells and ARNT in the HTR8 cells, HIF1α, HIF2α, and ARNT are transcribed in all of the studied cell lines when cultured under 20% O_2_ (Appendix A). These cell lines should all have the capacity to respond to hypoxic conditions using HIF-related mechanisms. Still, only the JEG3 and BeWo cells appeared to be sensitive to reduced oxygen concentrations, stimulating us to hypothesize that the presence of unidentified, additional, non-HIF-related transcriptional response mechanisms is specific to trophoblast cells that spontaneously express suppressyn. To further determine whether the oxygen concentration-associated *ERVH48-1* expression changes were mediated by hypoxia-induced changes in the HIF gene family members, pGL3 constructs lacking the HIF-binding sequence were created and assessed for their effects on the *ERVH48-1* promoter activities, again using luciferase assays. As predicted, decreased luciferase activities were seen for both the pGL3-LTR and pGL3-LTR-EIE constructs lacking an HIF-binding sequence (pGL3-LTR-HIFmg and pGL3-LTR-EIE-HIFmg, respectively) (Figure 5b, grey bar: vector-only transfected samples). 

Finally, we supported these results by examining the effects of 2% and 20% oxygen on the *ERVH48-1* promoter activity in cells transiently expressing HIF1α, HIF2α, and ARNT and combined HIF/ARNT complexes, all of which are intimately involved in the response to hypoxia in most cell types. The baseline transcription levels for these hypoxia-induced genes in cell lines cultured under 20% oxygen are shown in Appendix A. Note, in particular, that Ishikawa cells have a very low endogenous transcription of *HIF2α* and that HTR8 cells have a low endogenous transcription of *ARNT*. Since the baseline transcription under 20% oxygen conditions varied by cell line, we induced the transient expression of these genes in BeWo and HTR8 cells to confirm their effects on *ERVH48-1* promoter activity. The transient expression of HIF and ARNT translational products in BeWo cells was confirmed via an immunoblot analysis (Appendix A). In the BeWo cells, increases in the LTR activity were observed when HIF1α alone, HIF2α alone, and HIF1α/ARNT and HIF2α/ARNT complexes were induced. In contrast, LTR-EIE-associated luciferase activity increased only in the presence of HIF1α alone, HIF2α alone, or the HIF2α/ARNT complex. These results strongly support the hypothesis that the HIF1α and HIF2α proteins bind to the suppressyn promoter sequence and are involved in its increased expression in response to low oxygen levels (Figure 5b). The up-regulation of promoter activity under the transient expressions of HIF and ARNT protein family members was not observed for constructs lacking the HIF-binding sequence (-cacgt-) in either the LTR or the LTR-EIE. In contrast, HTR8 cells that do not express the suppressyn protein showed an induction of luciferase activities only in the LTR sequence that may bind the HIF1α/ARNT and HIFα/ARNT complexes, but its activities were disrupted for the LTR-EIE construct. In the HTR8 cells, no enhancer effect on the EIE sequence (pGL3-LTR-EIE) was noted with the expression of HIF alone, but a significant increase was observed with the expression of the HIF/ARNT complex when compared to the vector alone. However, this increase was lower than the expected enhancer effect of EIE (Figure 5c). The data from the HTR8 cells that endogenously express minimum ARNT gene products (Appendix A) indicate that ARNT is needed, in addition to HIFs, for LTR activation. These findings demonstrate that the HIF1α/ARNT and HIF2α/ARNT complexes bind to the *ERVH48-1* LTR sequence and that their presence is associated with significant enhancements in LTR expression, particularly in the presence of HIF2α/ARNT. We also noted an enhancement in *ERVH48-1* LTR’s hypoxia-induced promoter activities via the downstream sequence EIEs that are likely related to significant interactions between trophoblast-specific transcription factors (e.g., GATA family members) with the HIF2*α*/ARNT complex. It is notable, however, that although the EIE-related enhancement of the *ERVH48-1* LTR activities were preferentially associated with the presence of the HIF2α/ARNT complex, excess amounts (overexpressed) of ARNT dampens the enhancing effects of EIEs on both the HIF1α/ARNT and HIF2α/ARNT complexes (Figure 5b; pGL3-LTR-EIE). 

## 4. Discussion

Suppressyn, a protein that negatively regulates cytotrophoblast cell fusion into syncytiotrophoblast, is expressed almost exclusively in the placenta and is regulated by the surrounding oxygen concentrations [26]. To better understand the role of suppressyn in normal and abnormal human placental function, we sought to determine the transcriptional mechanisms underlying its tissue-specific expression and its regulation by oxygen. As suppressyn is derived from an integrated endogenous retrovirus sequence, *ERVH48-1*, we focused on the 5′ LTR that is common to all HERVs and examined its promoter potential. While the basic promoter activity for the *ERVH48-1* 5′ LTR sequence was confirmed in a variety of cell lines, irrespective of whether the specific cell line spontaneously produced suppressyn, we did not detect any of the enhancer sequences that are commonly found in most transcriptional regulatory sequences upstream of the *ERVH48-1* 5′ LTR. We therefore predicted a novel system for the tissue-specific transcriptional regulation of suppressyn expression.

Since suppressyn is specifically expressed in the placenta, we predicted that *ERVH48-1* would be susceptible to epigenetic modifications, including DNA methylation. Our results support the hypothesis that DNA methylation and histone modifications in the proximal promoter region of the 5′ LTR may be involved in mediating tissue (placenta)-specific suppressyn expression. Abnormalities in DNA methylation have been observed in the promoter region of the gene encoding syncytin-1, which is another endogenous retrovirus-derived protein that is expressed specifically in the placenta, in patients with various perinatal diseases [7,12,13]. Although there is no direct linkage published to date, we speculate that investigating potential DNA methylation abnormalities in *ERVH48-1* has the potential to identify links with diseases of abnormal placentation, possibly including pregnancy-induced hypertension, preeclampsia, fetal growth restriction, and recurrent early pregnancy loss, making future investigation in this area essential.

The in silico-generated data predicted enhancer sequences that were located within the *ERVH48-1* genome sequence and were designated as ERV Internal Enhancer sequences (EIEs). The EIEs exert strong enhancer effects in BeWo and JEG3 cells, supporting the hypothesis that they are placenta-specific, or at least suppressyn-specific enhancers. A further detailed analysis is needed, but GATA transcription factors, which are known to be important in placental development [4,40,41], are implicated here in the placenta-specific expression of suppressyn via binding to novel HERV-derived enhancer sequences within *ERVH48-1*. In summary, our newly described EIE sequences appear to act as transcriptional enhancer sites involved in the tissue-specific expression of suppressyn, and this may be mediated by placenta-specific GATA transcription factor binding.

An HIF-binding sequence can be identified in the 5′ LTR sequence of suppressyn. The results of the transient expression of a variety of HIF-associated genes in HTR8 cells strongly support the notion that the co-expression of ARNT with HIF1α or HIF2α is necessary for *ERVH48-1* LTR activation in response to low oxygen conditions. Further, only the trophoblast-derived cell lines that also endogenously express suppressyn (BeWo and JEG3 cells) displayed significant overall enhancer activities for the LTR-EIE construct, supporting the hypothesis that the EIE sequence has trophoblast-specific enhancer effects on hypoxia-induced LTR promoter activities. Note that specific responses to HIF2α/ARNT are higher than those associated with HIF1α/ARNT, which is consistent with HIF2α being more important in placental trophoblast-specific *ERVH48-1* gene expression. Endogenous ARNT protein, such as that present in BeWo cells, is sufficient for the hypoxia-induced gene regulation of *ERVH48-1* LTR activity, and the overexpression of ARNT in BeWo cells abrogates the enhancer effects of EIEs. Together, these results support the hypothesis that fine-tuning through the HIF2α/ARNT complex is the major regulator of suppressyn gene expression. 

The results of the experiments involving transient hypoxia-related gene transcription likely reflect immediate but not necessarily longer-term cell responses to activated HIF-related proteins. Since HIF1α and HIF2α protein levels and their related activities are controlled by the ubiquitin–proteasome systems, it is important that future investigations consider the intracellular degradation of HIF-related proteins and their dynamics in relation to the regulation of suppressyn gene expression [42]. 

We also detected several differences in response to low oxygen conditions between the trophoblast cell lines studied and primary trophoblast cells derived from term placentas. Although the overall responses to hypoxia were similar among all cell types, detected differences may be related to cell fusion potential and the state of cellular differentiation. To this point, previous studies have reported important roles of HIF1α and HIF2α in trophoblast syncytialization [43,44] and cytotrophoblast differentiation. In fact, a recent report showed that HIF2α plays a major role in normal placental development [45], a result that is consistent with our finding that the increased transcriptional activity of LTR-EIE was observed only for HIF2α in the BeWo cells. Since HIF1α and HIF2α genes have 48% homology, there may be a mechanism by which placenta-specific transcription factors and EIE sequences cooperate with HIF2α to regulate oxygen response. A further analysis of this possibility is warranted. Finally, it was suggested that the abnormally high nuclear expression of HIF may be involved in the development of pregnancy-induced hypertensive disorders and fetal growth restriction [45,46,47,48]. Our data would predict that the increased expression of HIF2α leads to the increased expression of suppressyn. As a potential central regulator of cytotrophoblast fusion/syncytiotrophoblast formation, the overzealous production and activity of suppressyn could result in abnormal placentation, a defect that is central to multiple common human pregnancy disorders. In short, the EIE-associated control of *ERVH48-1* may be necessary to dampen *ERVH48-1* LTR activities in the physiologically “hypoxic” placental microenvironment, as the overexpression of suppressyn in response to hypoxia could disrupt normal syncytialization.

## 5. Conclusions

In conclusion, the DNA methylation of promoter regions and novel placental-specific enhancer sequences (EIEs) regulate the tissue-specific expression of the gene encoding suppressyn (*ERVH48-1*), and the regulation of expression in trophoblast cells is strongly influenced by the expression of hypoxia-responsive transcription factors. These findings provide further support to the notion that the tightly controlled expression of suppressyn may be important for normal placentation and, conversely, that the high expression of HIF2α in response to hypoxia and/or abnormal DNA methylation of *ERVH48-1* promoter sequences may lead to the development of perinatal disease. These particular experiments will be the subjects of future work.

## Figures and Tables

**Figure 1 biomolecules-13-01627-f001:**
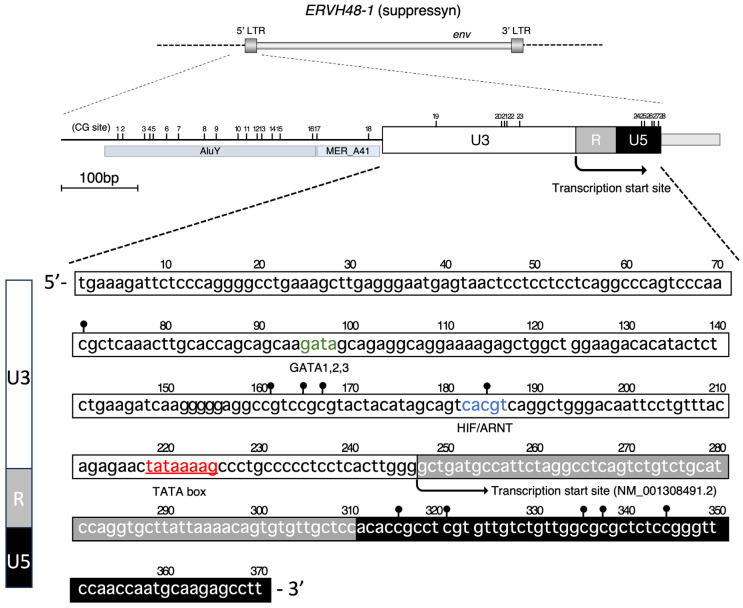
Schematic diagram of the suppressyn promoter region. The top panel shows the genome structure encoding suppressyn (*ERVH48-1*), and the middle panel shows an enlarged schematic of the 5′ LTR (long terminal repeat) and 5′ flanking regions. Lollipops indicate predicted CpG methylation sites. The lower panel shows the 5′ LTR sequence, with white outlining the U3, grey outlining the R, and black outlining the U5 regions. Arrows indicate the transcription start point of the NM001308491.2 transcript. Red lettering indicates TATA sequences, blue indicates predicted binding sequences for the HIF/ARNT complex, and green indicates predicted binding sequences for GATA transcription family members.

**Figure 2 biomolecules-13-01627-f002:**
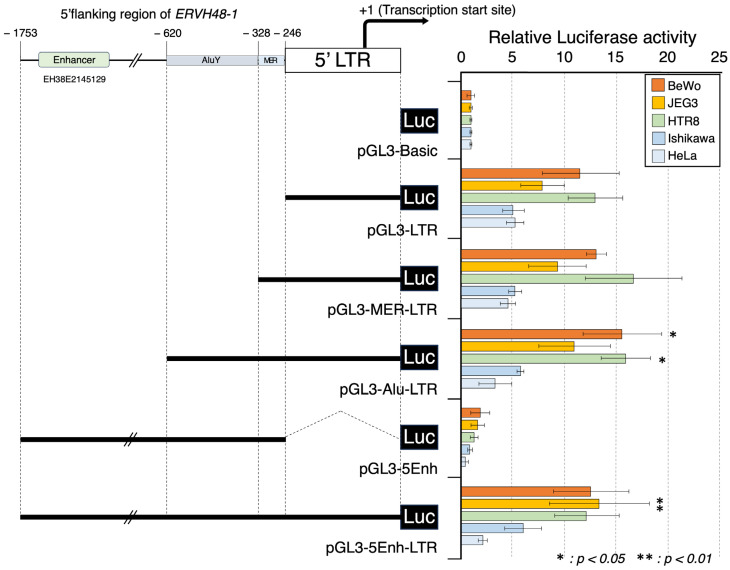
Promoter activity of the suppressyn 5′ LTR and its flanking region. On the left is a schematic diagram of the suppressyn 5′ LTR and 5′ flanking sequence, as cloned into the pGL3 vector. The enhancer region predicted by the cCRE is shown schematically (EH38E2145129). The right side of the diagram depicts the results of luciferase activities measured after the transfection of these constructs into various cell lines (BeWo, JEG3, HTR8, Ishikawa, and HeLa cells). pGL3-basic was used as reference for comparison to each of the other constructs. Statistical analyses were made using Mann–Whitney U tests compared to pGL3-LTR in each cell line (* *p* < 0.05, ** *p* < 0.01). Experiments were performed in duplicate, and the means and ± SDs were calculated from three independent experiments.

**Figure 3 biomolecules-13-01627-f003:**
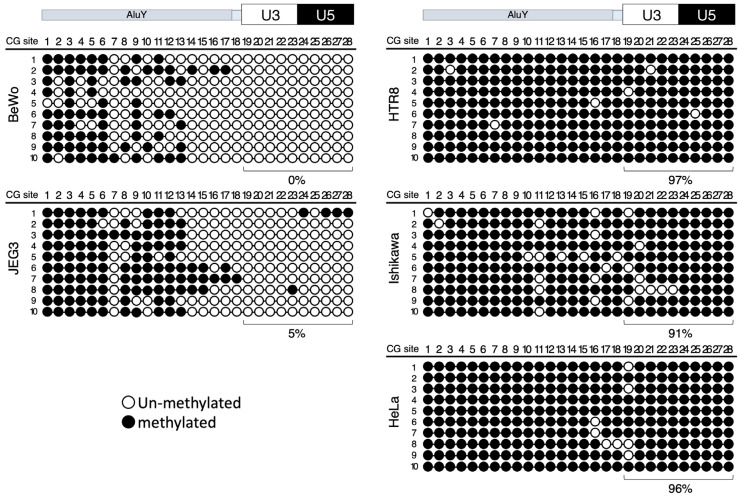
Bisulfite sequencing analysis of *ERVH48-1* promoter regions. Bisulfite sequencing of the *ERVH48-1* 5′ LTR and 5′ flanking region. White circles represent non-methylated CpG sites and black circles represent methylated CpG sites. The top columns show the location of CpG dinucleotides from 1 to 28, with numbers 19–28 depicting the methylation status of CpG sites within the 5′ LTR region. The bottom columns list the methylation rates (%) for each 5′ LTR region.

**Figure 4 biomolecules-13-01627-f004:**
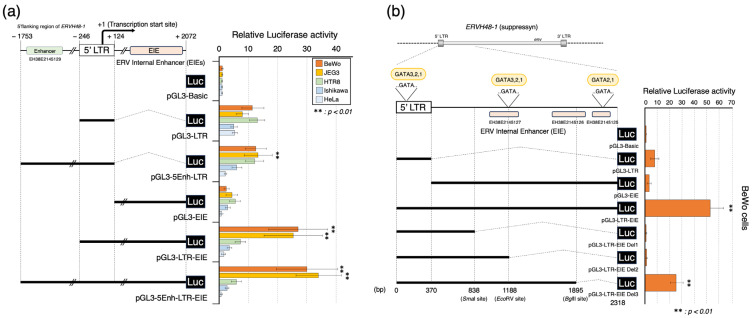
Luciferase activities of the *ERVH48-1* promoter and novel enhancer sequences (EIEs). (**a**) The schematic diagram on the left depicts the suppressyn 5′ LTR and its flanking sequences, including the 5′ flanking region of *ERVH48-1* and putative enhancer sequences (ERV Internal Enhancers; EIEs). Below, the separate constructs cloned into the pGL3 vector and the regions they cover, including the 5Enh, LTR, and EIE sequences, are shown. On the right, relative luciferase activity was plotted for these constructs for each cell line used (BeWo, JEG3, HTR8, Ishikawa, and HeLa cells). (**b**) The left side of this diagram depicts a schematic of the *ERVH48-1* 5′ LTR and EIE sequences. Details of the whole and the restriction enzyme-truncated (*Sma*I, *Eco*RV, and *Bgl*II) pGL3 constructs are shown below this. Three enhancer regions in the EIEs were predicted via cCRE. Predicted binding sites for GATA transcription factors in this region are presented (…GATA…). Luciferase activity associated with the transfection of these constructs into BeWo cells is shown on the right. Luciferase activity associated with the presence of the pGL3-basic construct was used as a comparator for statistical analysis. Mann–Whitney U tests were performed, and statistical significance was set at ** *p* < 0.01. Experimental replicates were performed in duplicate, and the means ± SDs were calculated from three independent experiments.

**Figure 5 biomolecules-13-01627-f005:**
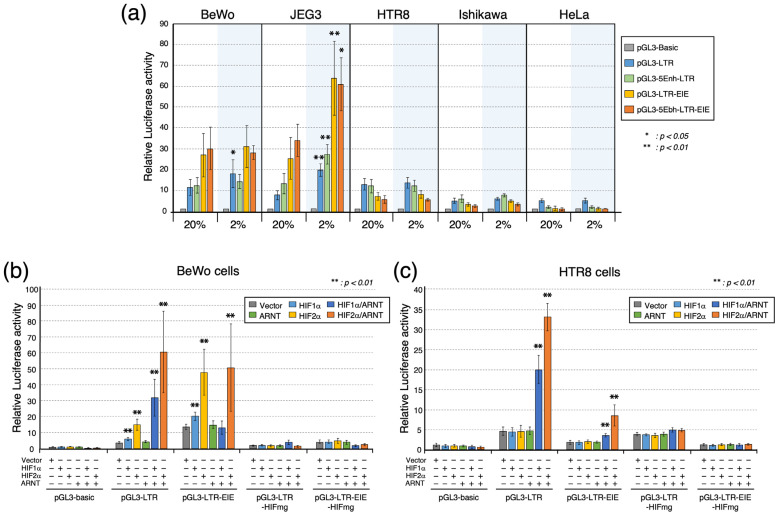
*ERVH48-1* promoter activity in low and high oxygen conditions. (**a**) Comparison of *ERVH48-1* promoter activity under low and high oxygen conditions (2% or 20%). Luciferase activity of *ERVH48-1* promoter sequences in BeWo, JEG3, HTR8, Ishikawa, and HeLa cells when cultured at 20% O_2_ (left column) or 2% O_2_ (right column). (Bars from left to right: pGL3-basic, -LTR, -5Enh-LTR, -LTR-EIE, and -5Enh-LTR-EIE.) Luciferase activities for each construct were normalized to pGL3-basic and compared to values for each reporter construct cultured under 20% O_2_ results to assess statistical differences. (Mann–Whitney U test: * *p* < 0.05; ** *p* < 0.01.) Experiments were performed in duplicate, and the means ± SDs were calculated from three independent experiments. (**b**) The effects of transient expression of hypoxia-inducible proteins (HIF1α, HIF2α, and ARNT) on the transcriptional activity of each *ERVH48-1* promoter construct in BeWo cells and (**c**) in HTR8 cells. pGL3-basic was included as a reference. Statistical analysis was performed, comparing the luciferase activities in cells exposed to the vector-only transfected control with those in which each gene or genes were transiently transfected (Mann–Whitney U test: ** *p* < 0.01). Experiments were performed in duplicate, and the means ± SDs were calculated from three independent experiments.

## Data Availability

Data supporting the findings of this study are available from the corresponding author upon reasonable request.

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
