# Peer review of "Controlling Trophoblast Cell Fusion in the Human Placenta—Transcriptional Regulation of Suppressyn, an Endogenous Inhibitor of Syncytin-1"

_biomolecules, 2023, doi:10.3390/biom13111627_

Round 1
Reviewer 1 Report
Comments and Suggestions for Authors
In this manuscript Sugimoto et al report analysis of DNA methylation and regulation of the ERVH48-1 (Suppressyn) LTR promoter region, and two novel predicted enhancer regions, including a downstream ERV Internal Enhancer (EIE). They demonstrate that the ERVH48-1 promoter is hypermethylated, and promoter-reporter activity of the ERVH48-1 LTR promoter, and the ERVH48-1 promoter-EIE enhancer, is low in non-trophoblast cell lines, whereas, in contrast, the LTR promoter is hypomethylated, and the promoter-EIE enhancer-reporter is active in trophoblast cell lines. They also demonstrate hypoxia treatment of trophoblast cell lines weakly enhances ERVH48-1 promoter- and more strongly enhances ERVH48-1 promoter-EIE enhancer-reporter activities, but not in non-trophoblast cell lines. The authors have identified evidence for GATA2 regulation via binding sites in the ERVH48-1 promoter- and ERVH48-1 promoter-EIE enhancers. In addition, evidence for HIF1a, HIF2a or ARNT involvement in hypoxia mediated regulation of the ERVH48-1 promoter-EIE enhancer in trophoblast cells was identified. Overall, this is an excellent study, although there are one or two minor points that I was curious about, or where clarification in the manuscript is needed, as follows:
1) Is there hypermethylation across the whole ERVH48-1 locus in non-trophoblast cells, such as HTR8, Ishikawa, or HeLa cells? Also, is there evidence for methylation differences in the EIE enhancer region between trophoblast versus non-trophoblast cells?
2) It would be of interest to additionally examine ERVH48-1 transcription in trophoblast and non-trophoblast cells under different oxygen levels, and/or high or low HIF1a, HIF2a, and ARNT levels. The authors suggested in line 540 that their data predict increased expression of HIF2a would lead to increased expression of suppressyn. Has this been investigated previously? Perhaps a comparison to investigations published previously, or a brief mention of what work is still needed, between ERVH48-1 promoter activity and ERVH48-1 mRNA expression under these conditions could be presented in the Discussion.
3) In line 179, the statement “…comprised of a U3 segment of at 246,..” does not make sense.
4) Supplementary figure 6 is presented in the supplementary figures, but is this referred to in the text?
5) In lines 531 and 535 and elsewhere, there is use of both HIF1a /HIF2a, and HIF-1a /HIF-2a – consistent use of terminology is needed.
6) Line 551, “These findings suggest provide…” grammar could be improved.
Reviewer 2 Report
Comments and Suggestions for Authors
This manuscript looked at suppressyn which is expressed in the placenta and its dysregulation may play a role in pregnancy complications associated with poor placentation. In this study, the authors aimed to “elucidate the mechanisms underlying suppressyn’s placenta-specific expression”. The experimental design in this study was unable to answer this question. In particular, the use of cell lines only, many of which are not appropriate models for studying the placenta, makes it very difficult to draw any significant conclusions from these findings.
Introduction
There are very few papers published on Suppressyn and the majority are from work from the authors of this manuscript however, there was no mention of the publication last year in Science from Frank et al DOI: 10.1126/science.abq7871 which showed that Suppressyn was expressed in the developing placenta. In this publication, the authors looked at the expression of Suppressyn in placenta tissue and cells by mining single-cell RNA sequencing data and showed expression of SUPYN was high in cytotrophoblasts and extravillous trophoblasts but was also present in syncytiotrophoblasts. Immunostaining also confirmed this in the second and third-trimester human placentas.
Minor changes
Please clarify DNA methylation throughout the introduction and lines 75-76 need clarification.
Results
Line 179 ‘comprised of a U3 segment of at 246’ please correct
Line 183 clarify DNA methylation
Line 222 clarify DNA methylation
Line 224 ‘tissue-specific expression of suppressyn’ how was this achieved? This is not possible with cell lines. Please refer to Frank et al 2022 Science DOI: 10.1126/science.abq7871
Lines 235-237 – There was no information about the samples used for this analysis. How many pancreas and thyroid samples were looked at? Need more than 1 to be able to claim levels of DNA methylation in these different tissues
Lines 238-241 ‘our results suggest that DNA methylation within the 5’LTR promoter region of ERVH48-1 appears to be a major regulatory mechanism ….’ This has not been shown by the findings in this study. Other epigenetic modifications including histone changes which have been shown by Frank et al 2022 Science are also important for the regulation of suppressyn and may be involved in tissue-specific expression.
Figure 3 – DNA methylation is shown but corresponding gene expression (RNA or protein) needs to be shown for these same samples
Line 251 – the 3 cCREs need to be shown on a figure
For the oxygen-related changes why were 2% and 20% chosen? Why wasn’t 8% investigated as 20% is very high? Also, these cells have always been cultured at 20% so are they the best way to test oxygen concentration changes?
Discussion
This is too long and a lot of the information in this discussion is covered in the Introduction or is a reiteration of the results (eg lines 375-381, 383-386, 407-409, 411-413, 432-437, 468-472)
Lines 426-429 – The authors should include DNA methylation analysis of placenta samples from preeclampsia or fetal growth restriction as this would strengthen the findings of this study and support the importance of DNA methylation in the regulation of suppressyn.
Reviewer 3 Report
Comments and Suggestions for Authors
The study presented by Sugimoto et al. focuses on the transcriptional regulation of the suppressyn gene.
Suppressyn, an endogenous retrovirus (ERVH48)-derived gene, was described several years ago by the same research team. The team systematically characterized this interesting gene, which was reported to inhibit the syncytialization process by binding to the receptor of sycyitin-1.
The current work focuses on the transcriptional regulation of the locus encoding suppressyn. The authors systematically analyze the genomic locus for promoter and potential enhancer activity. They conclude that the ERVH48-1 element functions as a promoter for the suppressyn gene. In addition, they identify a nearby sequence that has placenta-specific enhancer function for expression of the reporter gene. The authors also analyze expression regulation under different oxygen conditions.
This is a solid, descriptive work using classical assays (e.g., luciferase reporter assay, methylation assay, etc.) on various cultured cell lines. The experiments are clear. The Figures are informative.
Overall:
The quality of the work could be raised to a higher level by including more biologically relevant cells (not just cell lines) and additional molecular biology assays (e.g., ChIP-seq/ ChIP-qPCR, etc.), data mining of high-throughput data, single cells, and so on. For example, analysis of single cell expression data (at least data mining) would be highly informative, and would help to more accurately determine the cell type specificity of the enhancer.
In detail:
EIE (ERV Internal Enhancer) - The location of the genomic EIE region. Figures 1 and 4b show the ERVH48-1 element as an intact provirus containing intact open reading frames (ORFs) for gag pol and env. Does ERVH48-1 contain intact proviral ORFs? In the literature, this genomic locus is described as protein coding, but only for an env-like ORF. As shown in Figure 4b, the putative enhancer sequences of the EIE region overlap with the ORF gag. The figure needs to be corrected.
Binding sites for GATA transcription factors:
The authors claim to identify binding sites for GATA family transcription factors in the EIE region. However, based on the data shown, it is not convincing that the sequence motifs (...GATA...) identified as sequence motifs are actually binding sites for GATA transcription factors. Further validation experiments are needed to support such a conclusion (e.g. ChiP seq analysis (at least data mining), but ChiP qPCR analysis, site directed mutagenesis would be necessary.
Possible association with abnormal placentation and pregnancy:
The authors mention a possible link with disease: ".... Methylation abnormalities in ERVH48-1 may also lead to links with diseases with abnormal placentation, possibly including pregnancy-related hypertension, preeclampsia..." There are no reports in the literature of an association with abnormal placentation/pregnancy. Has suppressyn been mentioned in any disease association study (GWAS, etc.)? According to the literature, this particular locus is also mentioned as lncRNA and has been associated with cancer, but not pregnancy-related diseases.
Minor:
Lane 225
andJEG3 – correct: and JEG3
Lane 442
“… in HTR8, Ishikawa and HeLa cells, which 442 do not express suppressyn, the presence of these EIEs along with the 5_’ LTR promoter sequence exerted significant (p < 0.01) suppressive activity. While the mechanism for this effect remains unclear, we hypothesize that these EIEs disrupt transcriptional processes.”
It would be nice to explain what the authors have in mind. Discuss this rather mysterious transcriptional suppressive activity (if such) and cite literature, etc.
Lane 398
“These results suggest that transcriptional regulation of ERVH48-1 differs from that of the gene encoding syncytin-1 (ERVW-1), which is regulated by an enhancer sequence upstream of the promoter, even though both are HERV-derived genes characteristic of the human placenta [33].”
– Why do the authors think an enhancer should be upstream of the ERV?
Round 2
Reviewer 2 Report
Comments and Suggestions for Authors
Revisions are acceptable however the discussion is still extremely long and should refer to other studies on suppressyn.
Comments on the Quality of English LanguageThere have been many edits due to addressing reviewers comments but the manuscript needs to be proof read to correct small errors in spelling/grammar.
Author Response
Dear Reviewer,
We have made multiple changes to the manuscript. We feel we have specifically addressed the prior and most recent concerns of Reviewer 2. We have carefully proofed the entire manuscript to correct small errors in spelling/grammar as indicated in blue color. We have also added reference 4 to multiple relevant positions in introduction and discussion, as indicated in blue color to expand the references to other suppressyn-related publications. We have submitted a revised version with changes highlighted.